# LET IT CALM: EXPLORATORY ANNEALED DECODING FOR VERIFIABLE REINFORCEMENT LEARNING

## ABSTRACT

Reinforcement learning with verifiable rewards (RLVR) is a powerful paradigm for enhancing the reasoning capabilities of large language models (LLMs), yet its success hinges on effective exploration. An ideal exploration strategy must navigate two fundamental challenges: it must preserve sample quality while also ensuring training stability. While standard fixed-temperature sampling is simple, it struggles to balance these competing demands, as high temperatures degrade sample quality and low temperatures limit discovery. In this work, we propose a simpler and more effective strategy, Exploratory Annealed Decoding (EAD), grounded in the insight that exploration is most impactful on early tokens which define a sequence's semantic direction. EAD implements an intuitive *explore at the beginning, exploit at the end* strategy by annealing the sampling temperature from high to low during generation. This dynamic schedule encourages meaningful, high-level diversity at the start, then gradually lowers the temperature to preserve sample quality and keep the sampling distribution close to the target policy, which is essential for stable training. We demonstrate that EAD is a lightweight, plug-and-play method that significantly improves sample efficiency, consistently outperforming fixed-temperature sampling across various RLVR algorithms and model sizes. Our work suggests that aligning exploration with the natural dynamics of sequential generation offers a robust path to improving LLM reasoning.

## 1 INTRODUCTION

Reinforcement learning with verifiable rewards (RLVR) is a powerful approach to enhance the capabilities of Large Language Models (LLMs) in domains such as mathematical reasoning and code generation (OpenAI, 2024; Guo et al., 2025; Team et al., 2025; Yang et al., 2025). In this framework, an LLM learns by iteratively generating potential solutions (i.e., rollouts), and receiving feedback on its attempts. A central challenge lies in guiding language models to explore diverse yet high-quality solutions in their vast output space (Cheng et al., 2025); this reflects the long-standing hard trade-off between exploration and exploitation in RL (Thrun, 1992; Sutton et al., 1998).

To achieve effective exploration, the sampling process can be modified to increase the variance of its underlying distribution. However, any such modification involves two fundamental challenges. First, it must **preserve sample quality**. Increasing diversity at the cost of generating low-quality, nonsensical outputs is counterproductive. Second, it must **ensure training stability**. Modifying the sampler creates a discrepancy between the behavior policy (used for sampling) and the target policy (being optimized), which necessitates an importance sampling (IS) correction in the gradient update (Degris et al., 2012). If the probability ratio in the IS weight is too large, the gradients can have high variance, destabilizing the entire training process (Schulman et al., 2017). An ideal exploration technique must therefore increase diversity while keeping the sampling distribution close enough to the target policy to allow for stable learning (Haarnoja et al., 2018; Ziegler et al., 2019).

A widely adopted and principled way to this trade-off is to adjust the sampling temperature (Ackley et al., 1985; Hou et al., 2025). Beyond its implementation simplicity, this method is *variationally optimal*, that is, maximizing entropy (thereby increasing diversity) while bounding the KL divergence from the target policy (Jaynes, 1957). However, relying on a single fixed temperature creates a tension: high temperature promotes diversity but produces nonsensical text (Renze, 2024; Wang et al.,

2025b), whereas low temperature improves quality but limits exploration, leading to generic and repetitive outputs (Holtzman et al., 2020; Guo et al., 2025).

In this work, we propose **Exploratory Annealed Decoding (EAD)**, a strategy that improves the balance of this trade-off by leveraging a key insight into sequential generation: exploration is not equally valuable at every step. The initial tokens shape a sequence's semantic direction and structure, making early exploration crucial for discovering diverse valid solutions. Later tokens, however, fill in details within the established context, where excessive exploration can harm coherence. This insight motivates our core strategy: *explore at the beginning, exploit at the end*. This simple principle elegantly addresses the twin challenges of quality and stability. Injecting randomness early promotes diverse, high-level exploration, while reducing it later ensures completions are both coherent and close to the target policy—an essential property for stable off-policy learning.

In summary, our contributions are as follows:

① We propose EAD, a simple and effective exploration strategy for RLVR that dynamically anneals temperature to encourage meaningful diversity while maintaining high sample quality.

② We show EAD is a plug-and-play enhancement that improves sample efficiency over temperature sampling, delivering robust gains across various RLVR algorithms including GRPO (Shao et al., 2024), DAPO (Yu et al., 2025), and EntropyMech (Cui et al., 2025) on both small and larger models.

③ We show that EAD can be adapted for test-time inference, where a tuned temperature schedule further enhances generation quality.

## 2 PRELIMINARY

**Notations.** Let $x$ be a prompt from a dataset $\mathcal{D}$, and let $y = (y_1, \ldots, y_{|y|})$ be a generated response sequence, where $y_{<t}$ denotes the prefix $(y_1, \ldots, y_{t-1})$. We define an LLM as a policy $\pi_\theta$ parameterized by $\theta$, and denote the reference policy, i.e., the starting point for RL, as $\pi_{\text{ref}}$. The probability of generating $y$ given $x$ is defined autoregressively as $\pi_\theta(y \mid x) = \prod_{t=1}^{|y|} \pi_\theta(y_t \mid [x, y_{<t}])$. A reward model $R(x, y)$ evaluates the quality of a prompt-response pair; for our RLVR experiments, we use the rule-based Math-Verify reward model.[1] Finally, we use $|\cdot|$ to denote the length of a sequence or the cardinality of a set, and use the shorthand $1 : n$ for the set $\{1, \ldots, n\}$.

**Reinforcement Learning with Verifiable Rewards (RLVR).** The standard objective for Reinforcement fine-tuning of LLMs is to maximize the expected reward over a prompt dataset $\mathcal{D}$:

$$\max_\theta J(\theta) := \mathbb{E}_{x \sim \mathcal{D}, y \sim \pi_\theta(\cdot|x)} \left[ R(x, y) \right]. \tag{1}$$

However, on complex reasoning tasks, learned reward models $R(x, y)$ are prone to *reward hacking*, where they assign high scores to plausible but incorrect solutions (Gao et al., 2023; Perez et al., 2023; Weng, 2024; Wang et al., 2025a). RLVR addresses this by replacing the learned model with a verifiable, rule-based reward signal—such as a verifier that provides binary feedback on a solution's correctness (Guo et al., 2025). This ensures that the policy is optimized using a reliable signal.

RLVR is commonly implemented using policy gradient algorithms like Proximal Policy Optimization (PPO) (Schulman et al., 2017). However, standard PPO often requires complex token-level advantage estimation and a separate value model. To better suit RLVR's trajectory-level binary rewards, subsequent methods simplify the advantage calculation (Shao et al., 2024; Yu et al., 2025). A prominent example is Decoupled Clip and Dynamic Sampling Policy Optimization (DAPO) (Yu et al., 2025), which optimizes:

$$J_{\text{DAPO}}(\theta) =$$

$$\mathbb{E}_{x \sim \mathcal{D}, y^{(1:G)} \overset{iid}{\sim} \pi_{\theta_{\text{old}}}(\cdot|x)} \left[ \frac{1}{\sum_{i=1}^G |y^{(i)}|} \sum_{i=1}^G \sum_{t=1}^{|y^{(i)}|} \min \left\{ r_t^{(i)}(\theta) A_i, \text{clip} \left( r_t^{(i)}(\theta), 1 - \varepsilon_{\text{low}}, 1 + \varepsilon_{\text{high}} \right) A_i \right\} \right]$$

$$\text{s.t.} \quad 0 < \left| \{y^{(i)} : y^{(i)} \text{ is correct}\} \right| < G,$$

---

[1] https://github.com/huggingface/Math-Verify

where $\pi_{\theta_{old}}$ refers to previous policy and $r_t^{(i)}(\theta) = \frac{\pi_\theta(y_t^{(i)}|[x, y_{<t}^{(i)}])}{\pi_{\theta_{old}}(y_t^{(i)}|[x, y_{<t}^{(i)}])}$. The asymmetric bound $\varepsilon_{high} >$ $\varepsilon_{low}$ is proposed to relax the restriction on probability increase and encourage more exploration in the training. The advantage $A_i$ is computed by normalizing the binary rewards across a batch of $G$ responses, thus avoiding the need for a value model:

$$A_i = \frac{R_i - \text{mean}_{k \in 1:G}(R_k)}{\text{std}_{k \in 1:G}(R_k)}, \text{ where } R_i = R(x, y^{(i)}).$$

**Temperature.** Temperature sampling (Ackley et al., 1985) is a widely used method to control the stochasticity of the policy $\pi_\theta$. At each generation step $t$, the LLM computes a vector of logits, $\mathbf{h}$, over the vocabulary $V$ based on the prompt $x$ and the preceding tokens $y_{<t}$. Temperature sampling rescales these logits with a parameter $\tau > 0$ before applying the softmax function to form the next-token probability distribution:

$$\pi_\theta(y_t = v \mid [x, y_{<t}]; \tau) = \frac{\exp(h_v/\tau)}{\sum_{v' \in V} \exp(h_{v'}/\tau)},$$

where $v$ is a token in the vocabulary $V$ and $h_v$ is its corresponding logit. The temperature $\tau$ directly modulates the sharpness of the output distribution. A higher temperature ($\tau > 1$) flattens the distribution, increasing output diversity by making less likely tokens more probable. Conversely, a lower temperature ($\tau < 1$) sharpens it, leading to more deterministic, greedy outputs.

**Pass@$k$.** Pass@$k$ measures the probability that at least one of $k$ independent outputs from a language model is correct. Let $p_x$ denote the underlying accuracy of the language model $\pi$ given one prompt $x$. The pass@$k$ accuracy is defined as

$$\mathbb{E}_{x \sim \mathcal{D}} \left[1 - (1 - p_x)^k\right].$$

The inner term $1 - (1 - p_x)^k$ quickly approaches one unless $p_x$ is near zero. For example, when $k = 64$, any $p_x \geq 0.0695$ already yields a probability of at least $0.99$. A large gap between pass@$1$ and pass@$k$ suggests that for some prompts, $p_x$ is essentially zero. High diversity may benefit this metric because a model with diverse outputs is more likely to maintain non-negligible $p_x$ values across prompts, leading to stronger pass@$k$ performance.

**Entropy.** Given a prompt $x$ and a prefix $y_{<t}$, the **token-level entropy** at step $t$ is defined over the policy's conditional distribution:

$$H(Y_t \mid [x, y_{<t}]; \theta) := - \sum_{v \in V} \pi_\theta(y_t = v \mid [x, y_{<t}]) \log \pi_\theta(y_t = v \mid [x, y_{<t}]),$$

where the sum is over all tokens $v$ in the vocabulary $V$.

The **average entropy** of the policy, $H(\pi_\theta)$, is the expected token-level entropy over all prompts and generation steps. We can estimate this value empirically using Monte Carlo sampling. For each prompt $x \in \mathcal{D}$, we generate $G$ i.i.d. responses $y^{(1)}, \ldots, y^{(G)}$. The average entropy is then approximated by averaging the token-level entropies across all generated tokens:

$$\bar{H}(\pi_\theta) \approx \frac{1}{|\mathcal{D}|} \sum_{x \in \mathcal{D}} \left( \frac{\sum_{i=1}^G \sum_{t=1}^{|y^{(i)}|} H(Y_t \mid [x, y_{<t}^{(i)}]; \theta)}{\sum_{i=1}^G |y^{(i)}|} \right).$$

## 3 SEQUENTIAL EXPLORATION: EXPLORE EARLY, EXPLOIT LATE

Exploration is a cornerstone of reinforcement learning, enabling agents to discover high-quality policies rather than settling on suboptimal solutions (Sutton et al., 1998; Ladosz et al., 2022). This principle becomes particularly vital in deep RL, where vast action spaces render exhaustive search infeasible. In the context of RL for language models (e.g., RLVR), insufficient exploration often manifests as *entropy collapse*, i.e., a premature narrowing of the generation distribution during training (Yu et al., 2025; Wang et al., 2025b; Cui et al., 2025). A common simple tool to encourage exploration is *temperature sampling*. However, a *fixed* temperature imposes a difficult trade-off.

A high temperature promotes diversity (as indicated by increased entropy[2]), but it risks degrading output quality with nonsensical tokens and hallucinations (Renze, 2024; Wang et al., 2025b). In contrast, a low temperature limits the discovery of novel solutions, leading to generic and repetitive outputs (Holtzman et al., 2020; Guo et al., 2025).

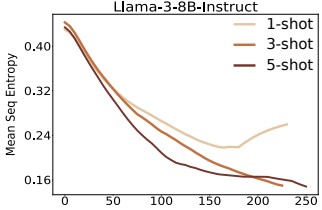

The key to resolving this dilemma lies not in finding a single best temperature, but in recognizing that exploration requirements vary throughout the generation process. This key insight stems directly from the autoregressive nature of language models. At the beginning of a sequence, the context is minimal and uncertainty is high, allowing a wide range of valid continuations. As more tokens are produced, the context becomes increasingly specific, constraining subsequent choices. We reference the Data Processing Inequality (DPI) (Shannon, 1948) as a *theoretical motivation* to investigate these entropy dynamics. While not a strict derivation for autoregressive models, the DPI provides a framework to hypothesize that expected conditional entropy tends to decrease as the context grows[3]:

Figure 1: Average entropy shrinks with output positions for Llama-3-8B-Instruct on MMLU dataset.

$$H(Y_t|[x, Y_{<t}]; \theta) = \underbrace{\mathbb{E}_{y_{<t}}\left[H(Y_t|[x, y_{<t}]; \theta)\right]}_{\substack{\text{Expected entropy at step } t \\ \text{(average over all prefixes } y_{<t})}} \geq \underbrace{\mathbb{E}_{y_{<t+1}}\left[H(Y_{t+1}|[x, y_{<t+1}]; \theta)\right]}_{\substack{\text{Expected entropy at step } t+1 \\ \text{(average over all prefixes } y_{<t+1})}} = H(Y_{t+1}|[x, Y_{<t+1}]; \theta)$$

We further validate this empirically by examining position-wise entropy trend on the MMLU dataset (Hendrycks et al., 2021)[4] with Llama-3-8B-Instruct (Grattafiori et al., 2024) (see Fig. 1).

This entropy decay has a direct performance implication: effective exploration, producing diverse, high-quality responses, is most beneficial in an uncertain, high-entropy phase at the start of generation. To verify this, we replicate a controlled "forking" experiment (Yang & Holtzman, 2025) on DeepSeek-Distilled Llama-3-8B (Guo et al., 2025), a representative RLVR-fine-tuned model from the same Llama-3 family. As shown in Fig. 2, trajectories branched from early generation steps consistently outperform those branched later. This finding is also consistent with observations from inference-time analysis, where forced, late-stage exploration tends to degrade output quality (Liao et al., 2025; Yang & Holtzman, 2025; Fu et al., 2025).

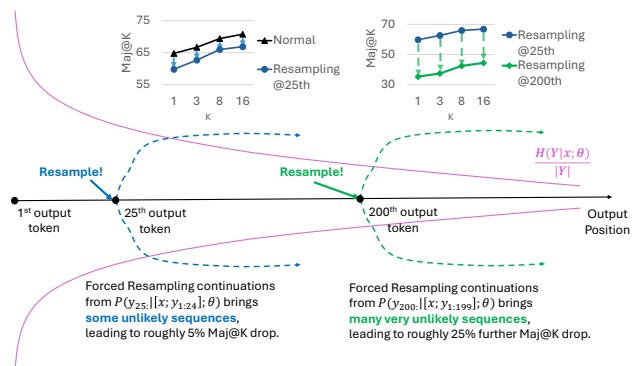

Figure 2: A "forking" experiment on DeepSeek-Llama-3 shows early branching (high-entropy region) yields higher Maj@$k$ on MMLU than late branching (low-entropy region).

Aligning our strategy with the natural dynamics of generation, we arrive at a simple yet powerful design principle: **explore early and exploit late**.

## 4 A METHOD FOR SEQUENTIAL EXPLORATION

To put the principle of "explore early, exploit late" into practice, we introduce *Exploratory Annealed Decoding (EAD)*, which uses an annealed temperature schedule starting from a higher-than-standard initial temperature (i.e., $\tau > 1$). To adapt this strategy to RLVR, we further incorporate a *global-step-aware decay rate*, ensuring that the temperature schedule remains effective as the typical response length increases during training.

---

[2]See Appendix C for the proof.

[3]While specific rollouts may have late high-entropy positions, the probability of this is exponentially small with position $t$ (Yang & Holtzman, 2025), making the overall trend a reliable heuristic.

[4]We use MMLU as a held-out dataset with Chain-of-Thought prompting (Wei et al., 2022) to incentivize longer reasoning outputs, aligning with a typical RLVR scenario.

**Exploratory Annealed Decoding.** Instead of a fixed temperature, our method dynamically adjusts the temperature $\tau_t$ for each token $t$ in a rollout. The schedule starts at a high temperature $\tau_{\max} > 1$ and decreases progressively throughout the generation process. Specifically, we sample the $t$-th token for one rollout with the token-level temperature $\tau_t = \max\{1 + \tau_{\max} - e^{t/d}, \tau_{\min}\}$, where we apply the annealed schedule with a *decay rate $d$* controlling the annealing speed.

As illustrated in Fig. 3, the decay rate $d$ controls how long the policy remains in a high-exploration state. A larger $d$ front-loads exploration across more initial tokens, while a smaller $d$ transitions to exploitation more quickly. In practice, we let $\tau_t = 1.0$ for $t < c$, where $c$ is a predetermined initial position for the sake of model-specific or prompt-specific template tokens injected in the training process. During RLVR, language models tend to generate some template tokens such as "*let's verify step by steps*" or repeat the question. We fix the temperature at $\tau = 1.0$ in this part to avoid interfering with the generation process.

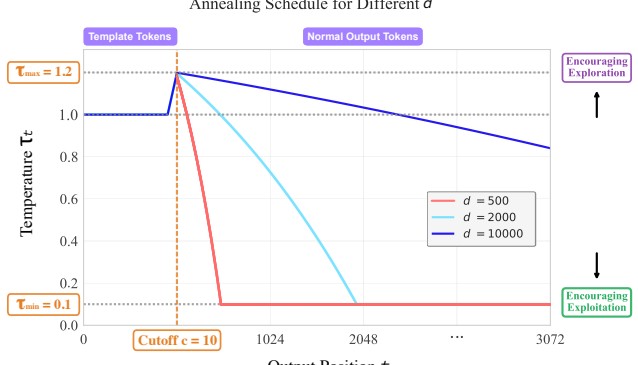

Figure 3: The annealing schedule with different decay rates $d$. A larger $d$ slows the cooling, front-loading exploration over more tokens. We set $c = 10, \tau_{\max} = 1.2, \tau_{\min} = 0.1$ for illustration.

**Global-Step-Aware Decay Rate.** As training progresses and response lengths increase,[5] the decay rate $d$ should be adjusted in accordance with the training step. Otherwise, an excessive number of tokens may be generated under extremely low temperatures, which degrades response quality and leads to undesirable behaviors such as repetition (Guo et al., 2025), off-topic drift (Spataru et al., 2024), and unnecessary verbosity (Holtzman et al., 2020). In particular, we adopt the following *global-step-aware decay rate*: $d_s = \min(d_0 + \alpha \times s, d_{\max})$, where $\alpha > 0$ is the growth factor and $d_{\max}$ is the decay cap.

**Ensuring Stability with Truncated Importance Sampling.** With aggressive annealing schedules (e.g., very small $\tau_{\min}$ and $d$), sampling low-probability, long-tail tokens can cause the annealed policy to deviate significantly from the one being optimized. This creates an off-policy discrepancy that risks training instability. To mitigate this, we employ *truncated importance sampling* (TIS) (Heckman et al., 1998; Hilton et al., 2022; Yao et al., 2025) to correct the objective, ensuring stable optimization even under highly exploratory schedules (see Appendix B for details).

Overall, this annealed decoding strategy offers a compelling combination of effectiveness and efficiency. As a plug-and-play modification to standard temperature sampling, it incurs negligible computational overhead and is fully compatible with existing RLVR pipelines and diverse policy optimization algorithms like DAPO, GRPO, etc.

## 5 EXPERIMENTS

### 5.1 EXPERIMENTAL SETUP

**Models, Data, and Training Frameworks.** To ensure a rigorous and controlled comparison, we follow the Minimal-RL recipe (Xiong et al., 2025),[6] training all models on the Numina-Math dataset (Beeching et al., 2024), which contains 860k math prompts. To assess the generality of our method, we experiment with both Qwen-2.5-Math-1.5B (Yang et al., 2024) and Llama-3.2-1B-Instruct (Dubey et al., 2024).[7] We also include the larger Qwen-2.5-Math-7B model to evaluate how

---

[5]We illustrate increased length for EAD in Appendix D.

[6]More training details and hyperparameter setups are illustrated in Appendix A.

[7]We also experimented with the Llama-3.2-1B base model. However, consistent with Wang et al. (2025d), we found that applying RL to base models without intermediate domain-specific fine-tuning yields limited gains across all methods. We defer a deeper investigation to future work.

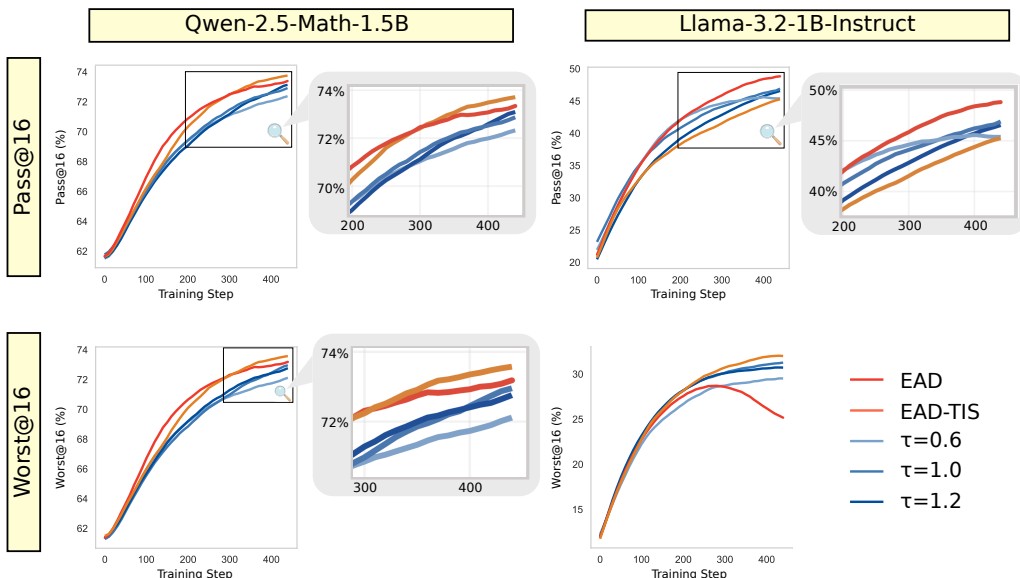

Figure 4: Pass@16 and Worst@16 performance evaluation in RL training. While EAD improves exploration of high-quality samples (even the worst outperform temperature sampling), the gain diminishes over time; importance sampling can supplement to correct bias and sustain training.

our approach scales. While our primary experiments are conducted within the DAPO framework (Yu et al., 2025), we demonstrate broader applicability by additionally integrating EAD with GRPO (Shao et al., 2024) and EntropyMech (Cui et al., 2025).

**Baselines and Controlled Comparison.** We evaluate EAD against fixed-temperature sampling, a standard and strong baseline, using temperatures $\tau \in \{0.6, 1.0, 1.2\}$ as recommended by prior work (Renze, 2024; Guo et al., 2025; Hou et al., 2025). For a fair comparison focused specifically on the sampling strategy, we disable two orthogonal techniques for all methods: (1) dynamic data sampling (Yu et al., 2025), to maintain a consistent training set for all runs, and (2) rollout length penalties, to avoid confounding the reward signal with length-based biases.

**Hyperparameters.** Unless otherwise stated, we use a default configuration of $\tau_{\max} = 1.2$, $d_0 = 25$, $\alpha = 5$, and $d_{\max} = 40000$ for EAD. For $\tau_{\min}$, we observed optimal values varied by model capability. For the 1B and 1.5B models, we set $\tau_{\min} = 0.1$. For the more capable 7B model, we used a higher value of $\tau_{\min} = 0.8$. All hyperparameters are tuned based on a prior study over held-out datasets.

## 5.2 EAD IMPROVES RLVR TRAINING

**EAD Improves RL Exploration and Training Efficiency.** As shown in Fig. 4, EAD significantly improves training efficiency. For Pass@16 accuracy, EAD (w/o TIS) consistently outperforms the baselines on the Llama and Qwen models (EAD (w/ TIS) also outperforms on the Llama model), demonstrating more effective exploration. Under the stricter Worst@16 metric, the inclusion of TIS becomes essential for maintaining stable performance gains, highlighting its importance for correcting the off-policy training dynamic introduced by EAD. Through bootstrapping evaluation as in Hochlehnert et al. (2025), the standard deviation of both Pass@16 performance and worst@16 are way below 0.01 and thus all comparisons here are significant.

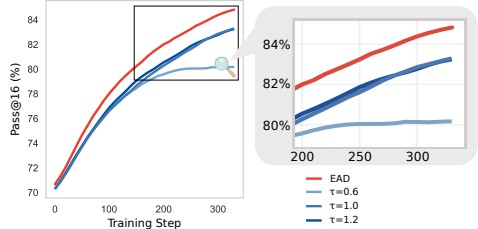

Figure 5: Pass@16 performance on Qwen-2.5-Math-7B. EAD enables better exploration than fixed-temperature sampling, yielding sustained gains in Pass@16 throughout training.

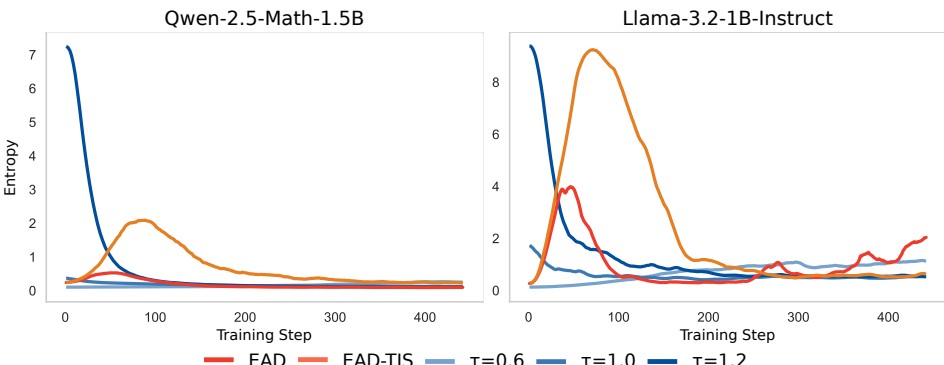

Figure 6: Entropy Dynamics in RL Training. Under commonly-used temperature sampling, trained with RL algorithm would make entropy decrease, sharply shrinking the exploration space for RL from beginning. While EAD could help RL algorithm to escape local minimum and do exploration when needed in the middle of RL training.

To verify that our method generalizes, we evaluated it on the larger Qwen-2.5-Math-7B model. The results, presented in Fig. 5, confirm that the performance gains from EAD remain significant. This demonstrates that our approach is effective not only on smaller models but also scales successfully.

**EAD Mitigates Entropy Collapse.** One major problem in RLVR training is entropy collapse (Cui et al., 2025), which causes the exploration space to shrink and constrains improvement during the "plateau stage" (Deng et al., 2025). We plot the entropy dynamic in Fig. 6, where we can see that the entropy dynamic for EAD-empowered methods is not monotonically decreasing from the beginning. Instead, it tries to gradually transition out from local optimum (Kirkpatrick et al., 1983; Bertsimas & Tsitsiklis, 1993) in a natural, continuous way without any external intervention, such as introducing tree search in rollout sampling (Li et al., 2025).

**Sample efficiency of EAD.** Increasing the number of rollouts is a common but computationally expensive strategy to enhance exploration (Hou et al., 2025). We test the sample efficiency of EAD by varying the number of rollouts, adjusting the learning rate accordingly as suggested by prior work (Chen et al., 2025). As shown in Fig. 7a, while more rollouts can further improve performance, EAD achieves strong results with just 4 or 8 rollouts. For instance, the optimal configurations are $n = 8$ with a learning rate of $10^{-6}$ for Llama-3.2-1B-Instruct and $2 \times 10^{-6}$ for Qwen-2.5-Math-1.5B. This highlights the sample efficiency of our approach, offering a way to reduce the computational cost of the rollout phase.

To assess whether EAD's advantage persists with extensive exploration, we compare it against baselines using a larger set of 16 rollouts. Fig. 7b shows that although the relative performance gain diminishes, EAD still outperforms fixed-temperature baselines by a clear margin.

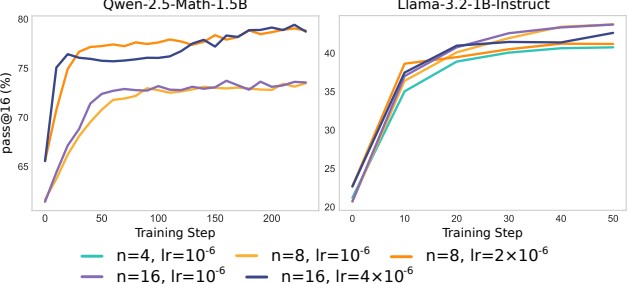

(a) EAD would bring further performance improvement via increased numbers of rollouts, but the commonly used 4 or 8 is already good enough.

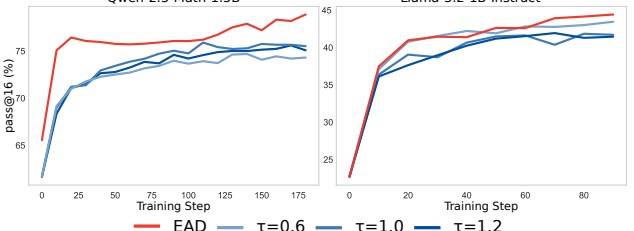

(b) When scaling out the rollout number to 16, the relative advantages of our methods diminished; however, it still outperforms traditional same-temperature sampling.

Figure 7: Sample Efficiency of EAD.

## 5.3 EAD IS COMPATIBLE WITH VARIOUS RL ALGORITHMS

To demonstrate that EAD is a general, plug-and-play exploration strategy, we evaluate its performance when integrated into two other prominent RL algorithms: GRPO (Shao et al., 2024) and EntropyMech (Cui et al., 2025). These algorithms provide diverse testbeds. GRPO is more conservative, constraining policy updates with a KL divergence penalty and stricter clipping mechanism that can limit exploration (Yu et al., 2025), while EntropyMech uses a specialized token-clipping mechanism to mitigate entropy collapse.

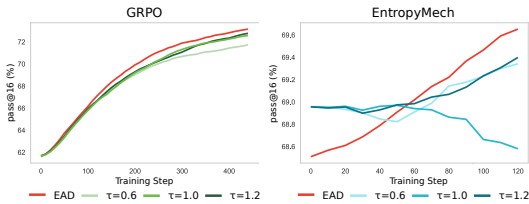

Figure 8: EAD is compatible with various RL algorithms and can significantly improve the model performance over time.

As shown in Fig. 8, EAD consistently outperforms fixed-temperature sampling in both frameworks. These results confirm the broad applicability of our method as an improved exploration strategy across different RL algorithms.

## 5.4 EAD IMPROVES INFERENCE-TIME SCALING

To understand whether the success of EAD in RL training is driven by its ability to generate high-quality samples, we conduct an evaluation at inference time. This experiment is designed to isolate the sampling strategy's effectiveness from the dynamics of RL optimization (Berseth, 2025). Using off-the-shelf Qwen-2.5 models without any fine-tuning, we compare EAD against fixed-temperature sampling. We use majority voting (Majority@$N$) to measure how performance scales with the number of samples $N$ (Wang et al., 2023; Snell et al., 2024). As shown

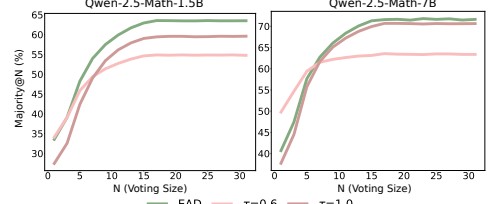

Figure 9: Inference-Time Scaling Evaluation for Different Decoding Methods using off-the-shelf Qwen2.5 models. We could see that EAD improves traditional temperature sampling. We set $\tau_{\max} = 1.2, \tau_{\min} = 0.1, d = 25$ for EAD.

in Fig. 9, EAD consistently improves over the baseline for most values of $N$. This result confirms that EAD's advantage stems from its inherent capacity to discover higher-quality solutions, even without any training.

## 5.5 ABLATION STUDY

**Reverse Annealing** We investigate a counter-intuitive baseline where the temperature increases during generation rather than decreasing. We employ the following reverse schedule: $\tau_t = \min\{\tau_{\min} + e^{t/d}, \tau_{\max}\}$. We term this schedule "RevEAD" and report its performance in Fig. 10. The results demonstrate that reverse annealing fails to even outperform standard temperature sampling, validating the necessity of an annealing (decreasing) temperature schedule.

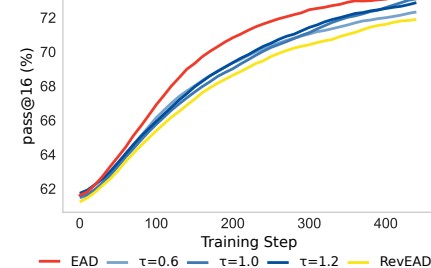

Figure 10: Reverse Annealing experiment results. The reverse schedule fails to surpass normal temperature sampling.

**Hyperparameter Sensitivity** We conduct an ablation study on the decay cap $d_{\max}$ and the growth factor $\alpha$. As shown in Fig. 11a, EAD is robust to a wide range of $d_{\max}$ values, provided the cap is sufficiently large to ensure a smooth annealing schedule. Conversely, a very small cap (e.g., $d_0 = d_{\max} = 25$) approximates a fixed schedule (equivalent to $\alpha = 0$ throughout training). This setting causes an abrupt performance drop, likely because the model cannot adapt to the increasing response lengths during training (see Appendix D). Alternatively, using a large initial $d_0$ yields a smooth schedule even with $\alpha = 0$. However, while training remains stable, performance improves slowly due to overly conservative exploration, as shown in Fig. 11b.

## 5.6 STRESS TEST: DAPO-17K RECIPE FOR AIME24

To verify the effectiveness of EAD on realistic mathematical reasoning tasks, we stress-test it using the DAPO recipe (Yu et al., 2025), training on DAPO-17k and evaluating on the challenging AIME 2024

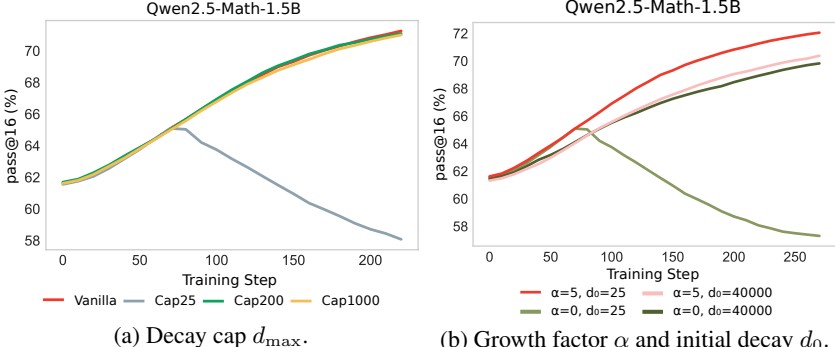

(a) Decay cap $d_{\max}$.  (b) Growth factor $\alpha$ and initial decay $d_0$.

Figure 11: Ablation on hyperparameter sensitivity. **Left:** Ablation on decay cap $d_{\max}$. "Vanilla" denotes EAD with the default $d_{\max} = 40,000$. Setting $d_{\max} = 25$ (where $d_0 = 25$) mimics a fixed temperature schedule (effectively $\alpha = 0$), leading to performance drops as the schedule fails to adapt to longer responses. EAD remains robust provided $d_{\max}$ allows for smooth temperature reduction (e.g., $d_{\max} \geq 200$). **Right:** Ablation on growth factor $\alpha$ and initial decay $d_0$. With a small $d_0$, a positive growth factor ($\alpha > 0$) is required to adapt to increasing response lengths and maintain a smooth intra-sequence schedule. While a large $d_0$ ensures smoothness even with $\alpha = 0$, it limits exploration, resulting in lower learning efficiency.

benchmark. To accommodate GPU memory constraints, we utilize the Qwen-3-8B-Base (Yang et al., 2025) model (instead of 32B) and reduce the maximum response length from 20,480 to 8,192. We set $d_0 = 500$ and $d_{\max} = 40,000$, and vary $\alpha \in \{5, 50\}$. As shown in Fig. 12, EAD achieves significant learning efficiency and consistently outperforms the standard temperature sampling baseline.

## 6 RELATED WORK

**Reinforcement Learning with Verifiable Rewards.** Recent large-scale reasoning models such as OpenAI o1 (OpenAI, 2024), DeepSeek-R1 (Guo et al., 2025) have demonstrated that reinforcement-learning-based post-training can substantially enhance LLM reasoning. Motivated by this reinforcement learning with verifiable rewards (RLVR) (e.g. Shao et al. (2024); Guo et al. (2025); Lambert et al. (2024); Yang et al. (2025); Hu et al. (2025); Yu et al. (2025); Guan et al. (2025); Zeng et al. (2025) among many other works) has become a major approach for post-training LLMs to improve reasoning. A broad literature studies how to make RLVR training effective and efficient at scale, including novel reinforcement learning algorithms and objectives (Yu et al., 2025; Liu et al., 2025b; Yue et al., 2025b;

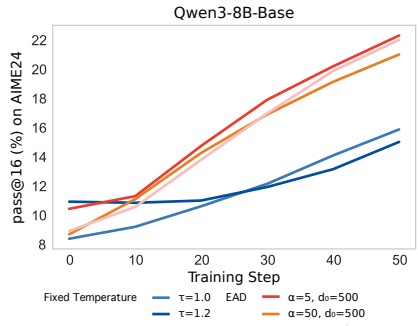

Figure 12: Stress-test of EAD using the DAPO recipe on AIME24. EAD outperforms the temperature sampling baseline across various hyperparameter setups.

Zheng et al., 2025a), verifier architecture and reward deigns (Zuo et al., 2025; Zhao et al., 2025b; Agarwal et al., 2025; Prabhudesai et al., 2025), and mechanisms that manage exploration diversity and entropy (Cheng et al., 2025; Chen et al., 2025; Cui et al., 2025; Wang et al., 2025b), or takes a critical view on the current evaluation (Yue et al., 2025a; Zhao et al., 2025a; Hochlehnert et al., 2025). Despite steady progress, a fundamental challenge is to balance exploration and exploitation along long reasoning trajectories without brittle heuristics. We adopt a simple yet effective annealed sampling schedule that front-loads exploration and cools later steps during rollout to encourage exploration while keeping training stable.

**Exploration Control in RLVR.** A line of works that is close to our work studies how to control exploration and sampling while doing RLVR (Hou et al., 2025; Cheng et al., 2025; Tang et al., 2025; Chen et al., 2025; Cui et al., 2025; Wang et al., 2025b; Xiong et al., 2025; Deng et al., 2025; Xu et al., 2025; Zheng et al., 2025b; Dou et al., 2025; Li et al., 2025). In particular, Cheng et al. (2025) propose per-token entropy to focus exploration at branching tokens in sampling; Tang et al. (2025); Chen et al. (2025) transform per-prompt rewards to optimize pass@$k$, guiding exploration across samples; Cui et al. (2025) control high-covariance tokens to prevent entropy collapse and sustain exploration; Wang et al. (2025b) update only high-entropy tokens, concentrating exploration where

decisions split. While more nuanced multi-threaded exploration strategies (Pan et al., 2025) could benefit training, adapting them to the training loop for large models often introduces significant computational overhead and the challenges of maintaining massive parallelization. Different from prior work, this paper presents the first systematic analysis of sampling temperature and introduce a purely sampling-level annealed schedule that encourages exploration and then progressively stabilize answers, thereby enabling discovery of new solutions while yielding more stable training.

**Simulated Annealing.** Simulated Annealing (SA) is a probabilistic optimization technique inspired by annealing in metallurgy, designed to find the global optimum in a large search space (Kirkpatrick et al., 1983; Bertsimas & Tsitsiklis, 1993). The core principle involves a temperature parameter that controls the probability of accepting suboptimal states. Initially, a high temperature allows the search to escape local minima by exploring broadly (exploration). As the temperature gradually decreases, the algorithm increasingly favors better states, converging towards a high-quality solution (exploitation). This "coarse-to-fine" search dynamic, where high temperatures establish a solution's general structure and low temperatures refine its details, strongly parallels the generative process of LLMs (Yang & Holtzman, 2025). SA has been adapted in various machine learning contexts to manage the exploration-exploitation trade-off, including recent applications in graph optimization (Liu et al., 2021), text editing (Zhang et al., 2024), non-autoregressive generation (Israel et al., 2025), and efficient Best-of-N sampling (Manvi et al., 2024). However, these prior applications invariably apply a single, uniform temperature across all positions in a generated sequence. This approach fails to account for the heterogeneous roles of tokens at different positions (Wang et al., 2025b). Our work departs from this convention. To the best of our knowledge, we are the first to introduce an **intra-sequence annealed temperature** schedule, where the temperature varies dynamically within the generation of a single sequence. This novel approach allows for more nuanced control over exploration and leads to significant performance gains in RLVR.

## 7 DISCUSSION

Our work addresses a central challenge in RLVR: achieve an effective balance between exploration and exploitation. We introduce Exploratory Annealed Decoding (EAD), a simple yet powerful sampling strategy that avoids heavy computation and intricate heuristics. Specifically, EAD employs a temperature-annealing schedule that begins with a high sampling temperature and gradually cools, enabling LLMs to explore broadly at the beginning of generation and converge toward precise, high-quality completions throughout the decoding process. As RLVR often relies on multiple rollouts to estimate rewards, this annealing schedule effectively improves sampling diversity while controlling variance, making it well suited for RL training. At the same time, EAD can also be applied directly at test time to enhance inference efficiency and scaling, improving the quality of single- or multi-sample decoding without additional computation cost.

Despite the encouraging results, our study has several limitations that suggest directions for future work. First, the scaling behavior of our method is not fully explored because of limited computational resources. We adopt the current settings with reference to (Xiong et al., 2025; Shao et al., 2025; Wang et al., 2025c), and argue that the efficacy of the method is still convincing, as we evaluate it across diverse model structures (LLaMA and Qwen) and multiple model sizes. A systematic scaling study remains an important next step. Second, while EAD is designed as a complementary component, a comprehensive study combining it with other advanced exploration-promoting RLVR algorithms (see § 6) remains a promising direction for future work. Third, our current experiments adopt a uniform temperature schedule for all prompts. Although an adaptive schedule tailored to individual prompts could potentially enhance performance, developing such a mechanism is nontrivial. In RLVR, training is iterative, so any prior information about prompt distributions may shift during optimization, and collecting extra statistics (e.g., token-wise entropy quantile (Wang et al., 2025b), or probability-advantage covariance (Cui et al., 2025)) to track these changes for every prompt would add computational overhead and system complexity (Li et al., 2025; Liu et al., 2025a). For these reasons, we focus on the vanilla schedule to test the core efficacy of our method, leaving adaptive scheduling for future investigation.

In summary, EAD provides a simple yet general way to couple exploration with the inherent progression of language generation. By reducing algorithmic overhead while improving trajectory quality, it opens new avenues for both efficient inference and effective reinforcement fine-tuning.

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

## A  MINIMAL-RL TRAINING DETAILS

We mainly follow the Minimal-RL recipe (Xiong et al., 2025) in our experiments to ensure a fair comparison among different rollout sampling strategies. Specifically, we set a series of hyperparameters as in Table 1:

| Hyperparameter | Value(s) |
|---|---|
| Training Batch Size | 1024 |
| Max Prompt Length | 1024 |
| Max Response Length | 3072 |
| Mini Batch Size | 256 |
| Micro Batch Size Per GPU | 4 |
| Learning Rate | $10^{-6}$ |

Table 1: Hyperparameter Setup for Running Minimal-RL recipe.

## B  OFF-POLICY ISSUE AND TRUNCATED IMPORTANCE SAMPLING CORRECTION

### B.1  SAMPLING TECHNIQUES CAN INTRODUCE OFF-POLICY ISSUE

One subtle yet troublesome drawback of reinforcement learning with sampling techniques is that it simultaneously introduces the *off-policy* problem: there is a gap between the behavior policy (used for sampling) and the target policy (being optimized and parametrized by $\theta$). This might introduce instability to the training and cause it to fail (See example training of RLVR with EAD Fig. 13).

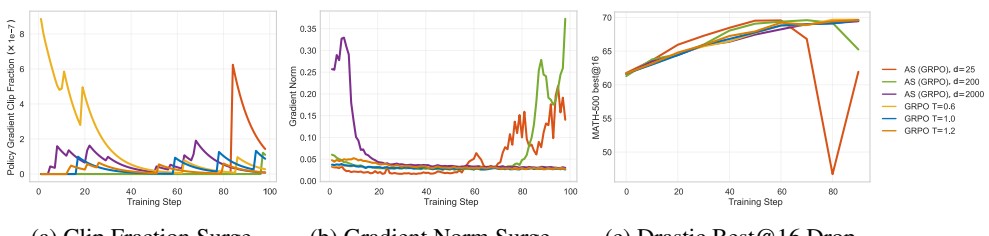

(a) Clip Fraction Surge      (b) Gradient Norm Surge.      (c) Drastic Best@16 Drop

Figure 13: Off-policy samples bring training instability. The base model is Qwen2.5-Math-1.5B.

Noted that this off-policy phenomenon widely exists for any efficient sampling framework (Yao et al., 2025) and sampling strategy (for instance, when applying best-of-$n$ sampling (Xiong et al., 2025) or filtering out responses (Shrivastava et al., 2025), the underlying distribution of responses is implicitly altered). To be more precise, we take the policy gradient loss as an example:

$$\mathbb{E}_{x\sim\mathcal{D},y\sim\pi_{\theta_{\text{old}}}^{\text{sampling}}(\cdot|x)}\left[\frac{\pi_\theta(y\mid x)}{\pi_{\theta_{\text{old}}}(y\mid x)}A(y;x)\right] = \mathbb{E}_{x\sim\mathcal{D},y\sim\pi_{\theta_{\text{old}}}(\cdot|x)}\left[\frac{\pi_{\theta_{\text{old}}}^{\text{sampling}}(y\mid x)}{\pi_{\theta_{\text{old}}}(y\mid x)} \times \frac{\pi_\theta(y\mid x)}{\pi_{\theta_{\text{old}}}(y\mid x)}A(y;x)\right],$$

where $\pi_{\theta_{\text{old}}}^{\text{sampling}}(\cdot\mid x)$ represents the underlying sampling distribution. In such case, an extra weight is implicitly added to each response in addition to its advantage $A(y;x)$.

This extra weight can significantly inflate the variance of the policy gradient, posing a stability challenge that our proposed EAD needs to mitigate. To quantify this effect in our proposed EAD, we now analyze such variance under a standard, fixed **temperature sampling**.

We use $\tau = 1$ to define a policy $\pi$ and consider the effect of $\tau$ on the variance of the gradient estimator. We reduce the problem to analyzing the variance inflation factor

$$\mathbb{E}_{y\sim\pi(\cdot|x;\tau)}\left[\frac{\pi(y\mid x;1)^2}{\pi(y\mid x;\tau)^2}\right]. \tag{2}$$

We begin with one-token case. Let $o_i$ denote the $i$th token in the vocabulary $V$ and $h_i$ is its logit. Then equation 2 can be rewritten as

$$\sum_{i=1}^{|V|} \frac{h_i/(\sum_{j=1}^{|V|} h_j)}{h_i^{1/\tau}/(\sum_{j=1}^{|V|} h_j^{1/\tau})} \times \frac{h_i}{\sum_{j=1}^{|V|} h_j} = \frac{\sum_{i=1}^{|V|} h_i^{2-1/\tau} \sum_{i=1}^{|V|} h_i^{1/\tau}}{\left(\sum_{i=1}^{|V|} h_i\right)^2}$$

**Proposition B.1.** *Suppose $h_i \in [0,1]$ for all $i \in V$. $\sum_{i=1}^{|V|} h_i^{2-1/\tau} \sum_{i=1}^{|V|} h_i^{1/\tau}$ is decreasing when $\tau \leq 1$ and increasing when $\tau \geq 1$, which implies it has a global minimum at $\tau = 1$.*

*Proof.* Let $x = 1/\tau$. We define

$$f(x) = \log\left(\sum_{i=1}^{|V|} h_i^{2-x}\right) + \log\left(\sum_{i=1}^{|V|} h_i^x\right).$$

Its derivative is

$$f'(x) = \frac{-\sum_{i=1}^{|V|} h_i^{2-x} \log h_i}{\sum_{i=1}^{|V|} h_i^{2-x}} + \frac{\sum_{i=1}^{|V|} h_i^x \log h_i}{\sum_{i=1}^{|V|} h_i^x}.$$

To analyze the sign of $f'(x)$, we define a helper function $g(x) = \frac{\sum_{i=1}^{|V|} h_i^x \log h_i}{\sum_{i=1}^{|V|} h_i^x}$. Then, $f'(x) = g(x) - g(2-x)$ and its sign depends on whether $g(x)$ is greater than, less than, or equal to $g(2-x)$. We take a look at derivative of $g$:

$$g'(x) = \frac{\left(\sum_{i=1}^{|V|} h_i^x (\log h_i)^2\right)\left(\sum_{i=1}^{|V|} h_i^x\right) - \left(\sum_{i=1}^{|V|} h_i^x \log h_i\right)^2}{\left(\sum_{i=1}^{|V|} h_i^x\right)^2} \geq 0.$$

Hence, $g$ is an increasing function and

$$\begin{cases} f'(x) = g(x) - g(2-x) \geq 0, & \text{when } x \geq 1 \\ f'(x) = g(x) - g(2-x) = 0, & \text{when } x = 1 \\ f'(x) = g(x) - g(2-x) \leq 0, & \text{when } x \leq 1. \end{cases}$$

Accordingly, $f$ is increasing when $x \geq 1$ and is decreasing when $x \leq 1$. Then the proposition easily follows. $\square$

The same conclusion can be proved for multiple-token sequence by induction. Therefore, we get that when the temperature is far from 1, the off-policy issue could be severe and lead to large variance of the gradient estimator.

## B.2 TRUNCATED IMPORTANCE SAMPLING RATIO CORRECTION

To cancel such bias, an importance sampling ratio can be introduced (Hilton et al., 2022; Yao et al., 2025):

$$\mathbb{E}_{x \sim \mathcal{D}, y \sim \pi_{\theta_{\text{old}}}(\cdot|x)}\left[\frac{\pi_\theta(y \mid x)}{\pi_{\theta_{\text{old}}}(y \mid x)} A(y;x)\right] = \mathbb{E}_{x \sim \mathcal{D}, y \sim \pi_{\theta_{\text{old}}}^{\text{sampling}}(\cdot|x)}\left[\frac{\pi_{\theta_{\text{old}}}(y \mid x)}{\pi_{\theta_{\text{old}}}^{\text{sampling}}(y \mid x)} \times \frac{\pi_\theta(y \mid x)}{\pi_{\theta_{\text{old}}}(y \mid x)} A(y;x)\right].$$

To further prevent negative effects by the extreme likelihood ratios and boost training stability, we truncate the likelihood ratio with an upper bound. That is, *truncated importance sampling* technique (Heckman et al., 1998). Taking the vanilla policy gradient loss as an example, the modified loss for EAD is as follows:

$$\mathbb{E}_{x \sim \mathcal{D}, y \sim \pi_{\theta_{\text{old}}}^{\text{EAD}}(\cdot|x)}\left[\min\left(\frac{\pi_{\theta_{\text{old}}}(y \mid x)}{\pi_{\theta_{\text{old}}}^{\text{EAD}}(y \mid x)}, \varepsilon\right) \frac{\pi_\theta(y \mid x)}{\pi_{\theta_{\text{old}}}(y \mid x)} A(y;x)\right].$$

## C    PROOF OF TEMPERATURE-ENTROPY RELATIONSHIP

**Proposition C.1.** *The entropy of the softmax distribution is a non-decreasing function of the temperature $\tau > 0$.*

*Proof.* The strategy is to show that the entropy $H$ is a non-increasing function of the inverse temperature $\beta = 1/\tau > 0$. The probability of sampling token $v$ with temperature $\tau$ is given by the policy $\pi_\theta$. For simplicity in the derivation, we denote this probability as $p_v(\tau)$:

$$p_v(\tau) \triangleq \pi_\theta(y_t = v \mid [x, y_{<t}]; \tau)$$

Let $h_v$ be the logit for a token $v$ in the vocabulary $V$. The probability of a token as a function of $\beta$ is given by:

$$p_v(\beta) = \frac{\exp(\beta h_v)}{\sum_{v' \in V} \exp(\beta h_{v'})} \triangleq \frac{\exp(\beta h_v)}{Z(\beta)},$$

where $Z(\beta)$ is the partition function. The entropy, as a function of $\beta$, is:

$$H(\beta) = -\sum_{v \in V} p_v(\beta) \log p_v(\beta).$$

We can rewrite the entropy by substituting $\log p_v(\beta) = \beta h_v - \log Z(\beta)$:

$$H(\beta) = -\sum_{v \in V} p_v(\beta)(\beta h_v - \log Z(\beta))$$

$$= \log Z(\beta) \left( \sum_{v \in V} p_v(\beta) \right) - \beta \sum_{v \in V} h_v p_v(\beta)$$

$$= \log Z(\beta) - \beta \cdot \mathbb{E}_{v \sim p(\beta)}[h_v].$$

Now, we differentiate $H(\beta)$ with respect to $\beta$. Let $\bar{h}(\beta) = \mathbb{E}[h_v]$.

$$\frac{dH}{d\beta} = \frac{d}{d\beta}(\log Z(\beta)) - \frac{d}{d\beta}(\beta \bar{h}(\beta)).$$

First, we find the derivative of the log-partition function:

$$\frac{d}{d\beta}(\log Z(\beta)) = \frac{Z'(\beta)}{Z(\beta)} = \frac{\sum_v h_v \exp(\beta h_v)}{Z(\beta)} = \sum_v h_v p_v(\beta) = \bar{h}(\beta).$$

Next, we use the product rule for the second term:

$$\frac{d}{d\beta}(\beta \bar{h}(\beta)) = \bar{h}(\beta) + \beta \frac{d\bar{h}}{d\beta}.$$

Combining these gives:

$$\frac{dH}{d\beta} = \bar{h}(\beta) - \left( \bar{h}(\beta) + \beta \frac{d\bar{h}}{d\beta} \right) = -\beta \frac{d\bar{h}}{d\beta}.$$

The derivative $\frac{d\bar{h}}{d\beta}$ is the variance of the logits. We can show this by differentiating $\bar{h}(\beta)$:

$$\frac{d\bar{h}}{d\beta} = \frac{d}{d\beta} \left( \frac{\sum_v h_v \exp(\beta h_v)}{Z(\beta)} \right)$$

$$= \frac{(\sum_v h_v^2 \exp(\beta h_v))Z(\beta) - (\sum_v h_v \exp(\beta h_v))Z'(\beta)}{Z(\beta)^2}$$

$$= \sum_v h_v^2 p_v(\beta) - \left( \sum_v h_v p_v(\beta) \right) \left( \frac{Z'(\beta)}{Z(\beta)} \right)$$

$$= \mathbb{E}[h^2] - (\mathbb{E}[h])^2 = \text{Var}_{v \sim p(\beta)}(h_v).$$

Substituting this back, we arrive at the final expression for the derivative of entropy:

$$\frac{dH}{d\beta} = -\beta \cdot \mathrm{Var}_{v \sim p(\beta)}(h_v).$$

By definition, the temperature $\tau > 0$, so the inverse temperature $\beta > 0$. The variance of any random variable is non-negative. This can be formally shown using **Jensen's inequality**: for the convex function $\phi(x) = x^2$, we have $\mathbb{E}[\phi(h)] \geq \phi(\mathbb{E}[h])$, which means $\mathbb{E}[h^2] \geq (\mathbb{E}[h])^2$, and thus $\mathrm{Var}(h) \geq 0$.

Therefore, the derivative of entropy with respect to $\beta$ is non-positive:

$$\frac{dH}{d\beta} = \underbrace{-\beta}_{\leq 0} \cdot \underbrace{\mathrm{Var}(h_v)}_{\geq 0} \leq 0.$$

Since $H(\beta)$ is a non-increasing function of $\beta$, and $\beta$ is inversely proportional to $T$, it follows that $H(\tau)$ must be a non-decreasing function of the temperature $\tau$. □

## D    INCREASING LENGTH IN RL TRAINING

During RL training, our algorithm (EAD) incentivizes the model to generate longer, more effective reasoning chains for difficult problems, especially for 7B models (Fig. 14). While both EAD and temperature sampling initially learn to shorten their responses by shifting from complex code-based solutions to direct mathematical reasoning, their behavior later diverges. The response length from temperature sampling stabilizes, whereas EAD learns to selectively increase reasoning length for harder problems, which boosts final performance. For these experiments, EAD is applied in the DAPO algorithm for sampling rollouts. We use the same training setup as detailed in § 5.

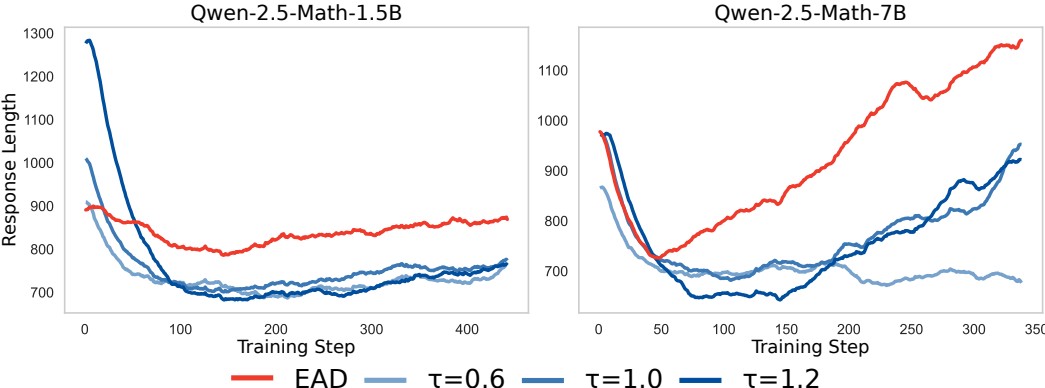

Figure 14: Compared with normal temperature sampling, EAD can naturally incentivize the model to generate longer reasoning chains.

