# OpenReview forum: "Let it Calm: Exploratory Annealed Decoding for Verifiable Reinforcement Learning"
_ICLR.cc/2026/Conference — Submitted to ICLR 2026_

### Official Review · Reviewer_xBao · 2025-10-28

**Soundness:** 3
**Presentation:** 4
**Contribution:** 2
**Rating:** 4
**Confidence:** 3

**Summary:**

The paper proposes an intuitively motivated strategy for controlling the exploration-exploitation trade-off during LLM generation for RLVR. Later on, they found this training strategy leads to many improvements on model performances in:
1. Mitigates entropy collapse
2. Improved Pass@k performance
3. Improves Majority@k performance during inference time

Overall I find this to be a solid, well executed, technical paper, though the idea is highly incremental. I think the paper meets the standard but is not necessarily the most exciting.

**Strengths:**

1. Clearly written paper
2. Significant Empirical Gains and Robustness.
3. Well measured benefits and good ablation studies

Exploration coefficient annealing has a long history in RL -- it's often known as epsilon-annealing in the epsilon-greedy exploration strategy. Modern LLMs exploration is controlled by temperature, therefore, annealing temperature intuitively will work.

**Weaknesses:**

1. Training Stability and Off-Policy Issues: The dynamic nature of EAD creates a discrepancy between the behavior policy and the target policy, introducing an off-policy issue that risks training instability.
2. Hyperparameter Sensitivity: The optimal configuration of EAD depends on model capability.
3. Diminished Relative Gains at High Exploration: While EAD remains superior, the relative performance advantage diminishes when the number of rollouts is scaled up extensively.

**Questions:**

The benefits of temperature annealing on pass@k, majority@k (inference time) is more like an **after-the-fact realization**. If inference-time scaling is all I care about, then I could try other methods, such as in [1]. They directly proposed policy optimization algorithm using pass@k as reward. How does temperature annealing, a more general method, compare to those that specifically made to target pass@k?

I know it might be difficult to add a new experiment during rebuttal -- so I don't want to force too much work on the authors. But if they resubmit the paper to another venue, they should consider adding some comparison studies to show that a general method such as EAD can even outperform methods specifically designed for inference-time scaling. That would make this paper's result extremely surprising and exciting since EAD is in an order of magnitude simpler to implement than method proposed in [1]. However, without the comparison, I have no idea which one to use.

[1] Tang, Yunhao, et al. "Optimizing language models for inference time objectives using reinforcement learning." arXiv preprint arXiv:2503.19595 (2025).

---

> ### Author Response · Authors · 2025-11-21
> **Response to Weaknesses**
>
> We thank Reviewer xBao for their constructive feedback and for recognizing the clarity and effectiveness of our proposed EAD methods. We particularly appreciate the connection drawn to traditional RL literature; revisiting these "test-of-time" techniques in the context of modern LLM RLVR is indeed a core motivation of our work.
>
> > **W1**: Training Stability and Off-Policy Issues: The dynamic nature of EAD creates a discrepancy between the behavior policy and the target policy, introducing an off-policy issue that risks training instability.
>
> **R1**: This is an excellent point. We acknowledge that annealing the temperature theoretically widens the gap between the sampling policy and the target policy. As discussed in Section 4 and Appendix B, we mitigate this by employing classical Truncated Importance Sampling (TIS) to prevent unbounded importance ratios.
> Empirically, we do not observe severe performance degradation (Figures 4 and 5). On the contrary, the EAD temperature schedule effectively mitigates the "entropy collapse" phenomenon [1, 2], allowing the model to continuously improve. We argue that the benefit of preventing collapse outweighs the theoretical off-policy risks in this context. We agree that further theoretical analysis of this trade-off is a promising direction for future work.
>
> > **W2**: Hyperparameter Sensitivity: The optimal configuration of EAD depends on model capability.
>
> R2: In this work, we tune hyperparameters on the validation set. Generally, EAD proves robust across varying task difficulties. Our ablation studies show that decay rates between 25 and 2000 are effective. Specifically, larger rates (e.g., 500) benefit harder tasks like AIME-24 (requiring longer exploration horizons), while lower rates (e.g., 25) suffice for simpler tasks like MATH-500.
>
> > **W3**: Diminished Relative Gains at High Exploration: While EAD remains superior, the relative performance advantage diminishes when the number of rollouts is scaled up extensively.
>
> **R3**: We view this behavior as a demonstration of EAD’s sample efficiency. While scaling up rollouts is a common strategy for exploration [3], it incurs linear increases in computational cost (throughput reduction and memory footprint). In practical settings with limited compute budgets, methods that maximize exploration quality per sample are preferred. As shown in Figure 7 (Section 5.2), EAD achieves significant performance gains using only 4 or 8 rollouts, capturing the majority of the benefit with minimal computational overhead compared to massive rollout strategies.

---

> ### Author Response · Authors · 2025-11-21
> **Responses to Questions**
>
> > **Q1**: The benefits of temperature annealing on pass@k, majority@k (inference time) is more like an after-the-fact realization. If inference-time scaling is all I care about, then I could try other methods, such as in [4]. They directly proposed policy optimization algorithm using pass@k as reward. How does temperature annealing, a more general method, compare to those that specifically made to target pass@k?
>
> **A1**: We emphasize that sampling strategies (like EAD) and training objectives (like pass@k optimization) are orthogonal and complementary components of RL training. EAD is a "plug-and-play" sampling strategy that can be applied regardless of the specific loss function or reward structure. As demonstrated in Section 5.4, we successfully combined EAD with GRPO and EntropyMech to achieve additive performance gains. Practitioners do not need to choose between EAD and specific policy optimization algorithms; EAD simply ensures better exploration during the training of those algorithms.
>
> > **Q2**: I know it might be difficult to add a new experiment during rebuttal -- so I don't want to force too much work on the authors. But if they resubmit the paper to another venue, they should consider adding some comparison studies to show that a general method such as EAD can even outperform methods specifically designed for inference-time scaling...
>
> **A2**: We appreciate the reviewer’s understanding regarding the timeline and the insightful suggestion. To clarify, EAD is primarily designed to improve the exploration-exploitation trade-off during RL training, yielding a better reasoning model. The inference-time benefits shown in Section 5.3 serve to further validate the robustness of the method.
>
> However, we reiterate our position from A1: comparisons with methods like [4] are valuable, but ultimately these approaches are complementary. EAD improves the data distribution the model learns from, while methods like [4] optimize the objective. We will clarify this distinction and the potential for combined usage in the revision.
>
>
> References:
>
> [1] Cui, Ganqu, et al. "The entropy mechanism of reinforcement learning for reasoning language models." arXiv preprint arXiv:2505.22617 (2025).
>
> [2] Wu, Fang, et al. "The Invisible Leash: Why RLVR May or May Not Escape Its Origin." arXiv preprint arXiv:2507.14843 (2025).
>
> [3] Hou, Zhenyu, et al. "T1: Advancing Language Model Reasoning through Reinforcement Learning and Inference Scaling." Forty-second International Conference on Machine Learning.
>
> [4] Tang, Yunhao, et al. "Optimizing language models for inference time objectives using reinforcement learning." arXiv preprint arXiv:2503.19595 (2025).

---

### Official Review · Reviewer_fG7W · 2025-10-31

**Soundness:** 3
**Presentation:** 3
**Contribution:** 1
**Rating:** 2
**Confidence:** 5

**Summary:**

This paper introduces Exploratory Annealed Decoding (EAD), a novel sampling strategy for Reinforcement Learning with Verifiable Rewards (RLVR). The method is motivated by the observation that the conditional entropy of a language model's output distribution naturally decreases as a sequence is generated. Grounded in this insight, EAD proposes an "explore at the beginning, exploit at the end" strategy by dynamically annealing the sampling temperature from a high value to a low value during the generation of a single sequence. This approach aims to encourage high-level semantic diversity in the early tokens, which are argued to be most critical for a sequence's overall direction, while ensuring coherence and stability by reducing randomness in later tokens.

**Strengths:**

The paper is well-written and easy to follow. The motivation is clearly articulated, grounding the proposed method in an intuitive principle ("explore early, exploit late") and supporting it with theoretical reasoning and empirical evidence.

**Weaknesses:**

The paper's contributions are undermined by its conceptual simplicity and an over-generalized central assumption.

1. Limited Novelty: The core idea of decreasing temperature over time to balance exploration and exploitation is the foundational principle of Simulated Annealing, a classic optimization algorithm. While the authors correctly note that they are the first to apply this intra-sequence, this adaptation feels more like an incremental application of a well-known heuristic rather than a fundamentally new technique. The conceptual leap from annealing over optimization steps to annealing over generation steps in a sequence is not substantial enough to be considered a major contribution on its own.

2. Overly Simplistic and Brittle Heuristic: The central premise that exploration is most valuable for early tokens is a strong assumption that does not hold universally, particularly in the complex reasoning tasks that RLVR targets. Many reasoning processes require precision and the discovery of critical, low-probability tokens late in the sequence. For example:

i) In a multi-step mathematical proof, the final numerical answer or a key simplification step might occur near the end. A very low temperature at this point would suppress the model's ability to find the correct, precise token if it's not the most probable one.

ii) In code generation, a bug might be fixed by changing a single character or keyword in the final line of a function. Early exploration might set the overall structure, but late-stage exploration is needed to find that specific, crucial fix.

In general, the "explore early, exploit late" strategy seems brittle and tailored to problems where the initial plan is all that matters, which is a limiting assumption for general-purpose reasoning. Other works have proposed more nuanced exploration strategies, such as focusing on high-entropy "branching" tokens regardless of their position, which seems like a more robust and principled approach (e.g. https://arxiv.org/abs/2504.15466)

**Questions:**

1. The paper's core heuristic seems to break down for reasoning tasks that require high precision or discovery in later generation steps. Could you discuss the failure modes of EAD? Have you analyzed its performance on tasks where a single, critical token late in the sequence determines correctness (e.g., solving an equation for a final numerical value)?

2. The primary baseline is fixed-temperature sampling. While a standard choice, it doesn't fully challenge the dynamic nature of EAD. Have you considered comparing EAD to other dynamic schedules? For example, a "reverse" annealing schedule (exploit then explore) or a schedule that adapts the temperature based on the model's current uncertainty (e.g., increasing temperature when the next-token entropy is high), irrespective of position?

3. The global-step-aware decay rate appears to be a hand-tuned heuristic (line 240). Could you provide more justification for this specific functional form? An ablation study on the sensitivity of the linear growth factor (5) and the cap (40000) would strengthen the claim that EAD is a robust, "plug-and-play" method.

4. You mention that for the 7B model, a higher \tau_min was needed to prevent it from generating "plausible but incorrect solutions." This seems to contradict the core motivation of exploiting (i.e., generating high-probability, high-quality tokens) at the end. Could you elaborate on this? Does this suggest that for more capable models, a low temperature late in the sequence is actually harmful, thus undermining the "exploit late" principle?

---

> ### Author Response · Authors · 2025-11-21
> **Responses to Weaknesses**
>
> Thanks for suggesting the related work. However, we respectfully disagree that our contribution should be regarded as incremental. We believe that applying a simple yet effective idea to an important new use case (RLVR) with strong empirical results represents a meaningful contribution. Specifically, we highlight the following points:
>
> 1. **EAD is a novel and lightweight training strategy with careful design**. Below we summarize the key design principles and clarify points that may have been misunderstood.
>      - **Sample efficiency and effective exploration**. We argue that early-stage branching exhibits its unique advantage that it promotes diversity of rollouts and hence is beneficial to exploration. In contrast, later-stage branching provides trajectories with the same prefix offering limited additional information for training. In this sense, we think early-stage branching is more beneficial to effective exploration.
>
>     - **Training stability**. Our method is designed to eliminate reliance on high temperature at the final step and thus mitigates off-policy issues in RL training. More specifically, if a model requires high entropy at the final step to "stumble" upon the solution (i.e., sampling a low-probability token, as low-temperature won’t miss high-probability candidates), the reasoning trajectory might hurt the training stability. This is because training on such "lucky" successes creates high variance and an off-policy issue (discussed in App. B). On the other hand, our annealing schedule forces the model to "commit" to its reasoning early. By reducing temperature later, we ensure the model learns robust paths where the correct answer follows logically from the prefix, rather than from random sampling.
>
>     - **Scalability and system complexity**. We kindly point out that infrastructure complexity is a real concern in practice and EAD adds zero-overhead. In contrast, nuanced strategies may require substantial engineering, including multi-threading, message-passing synchronization, and tool-calling formulations as stated in [1]. This complexity may limit its scalability. Notably, [1] conducts experiments on small 228M models, which is much smaller than standard LLM scales (e.g., 1B/7B+). Given this gap, we believe that a simple, easily implemented method that yields consistent empirical gains is valuable in practice.
>
>     - **Orthogonality with other approaches**. EAD is actually orthogonal to all inference-time branching techniques (such as [1]) and other training algorithms. Its plug-and-play nature allows it to be easily combined with other techniques, and we encourage such integration. In the paper, we combine it with multiple training recipes.
>
> 2. **The value of simplicity**: While our approach is simple, this should not be conflated with being incremental. First, EAD represents a non-trivial adaptation of the widely-recognized principle of simulated annealing to modern RLVR scenarios, supported by concrete empirical observations of entropy dynamics (Section 3, also related work [2,3]) and a novel solution to the off-policy problem specific to RLVR (Section 4). Second, the simplicity is by design: it ensures practical ease of implementation and computational efficiency with zero additional overhead (e.g., no complex system design and no additional statistics like entropy to track). Third, despite the simplicity, our comprehensive experiments in Section 5 demonstrate consistent improvements across diverse training objectives (DAPO, GRPO, EntropyMechanisms) and recipes (Minimal RL and DAPO-17K). We believe this contribution encourages broader consideration of rollout sampling strategies in RLVR.
>
> In the revised manuscript, we now explicitly emphasize efficiency, effectiveness, and flexibility as key advantages of our design, and distinguish our contribution with prior work.
>
>
> **References**:
>
> [1] Pan, Jiayi, et al. "Learning adaptive parallel reasoning with language models." arXiv preprint arXiv:2504.15466 (2025).
>
> [2] Wang, S., Yu, L., Gao, C., Zheng, C., Liu, S., Lu, R., Dang, K., Chen, X., Yang, J., Zhang, Z. and Liu, Y., 2025. Beyond the 80/20 rule: High-entropy minority tokens drive effective reinforcement learning for LLM reasoning. arXiv preprint arXiv:2506.01939.
>
> [3] Cui, G., Zhang, Y., Chen, J., Yuan, L., Wang, Z., Zuo, Y., Li, H., Fan, Y., Chen, H., Chen, W. and Liu, Z., 2025. The entropy mechanism of reinforcement learning for reasoning language models. arXiv preprint arXiv:2505.22617.

---

> ### Author Response · Authors · 2025-11-21
> **Responses to Questions**
>
> > **Q1**: The paper's core heuristic seems to break down for reasoning tasks that require high precision or discovery in later generation steps. Could you discuss the failure modes of EAD? Have you analyzed its performance on tasks where a single, critical token late in the sequence determines correctness (e.g., solving an equation for a final numerical value)?
>
> **A1**:  Please refer to our response to the Weaknesses for the critical token case ("Training Stability").
> One failure mode arises when the decay rate $d$ is too small, especially when the ending temperature is low. We need to avoid the case where an excessive number of tokens are generated at an extremely low temperature to avoid repetition, off-topic drift, and unnecessary verbosity, etc. We previously mentioned this in Section 4, paragraph “Global-Step-Aware Decay Rate” in the original manuscript.
> In the revised version, we’ve added additional ablation studies on the hyperparameters, with results presented in Figure 11 and discussed in Section 5.5 in the revision. These new experiments further confirm that the decay rate cannot be too small throughout the training. Translating this principle into practical guidance for hyperparameter selection: (1) the decay cap $d_\max$ should not be too small (need $d_\max\ge200$), and (2) if the initial decay rate $d_0$ is small, a positive growth factor $\alpha$ is required.
>
> > **Q2**: The primary baseline is fixed-temperature sampling. While a standard choice, it doesn't fully challenge the dynamic nature of EAD. Have you considered comparing EAD to other dynamic schedules? For example, a "reverse" annealing schedule (exploit then explore) or a schedule that adapts the temperature based on the model's current uncertainty (e.g., increasing temperature when the next-token entropy is high), irrespective of position?
>
> **A2**: Thanks for raising this interesting point. We’ve added an ablation study to test the efficacy of the “reverse annealing schedule” in Section 5.5 in the latest version. The training dynamics of EAD, vanilla temperature sampling, and reverse annealing are shown in Figure 10. As shown, the yellow line corresponding to reverse annealing exhibits the worst performance. This observation further supports that the training dynamics of EAD are more natural and beneficial.
>
> > **Q3**: The global-step-aware decay rate appears to be a hand-tuned heuristic (line 240). Could you provide more justification for this specific functional form? An ablation study on the sensitivity of the linear growth factor (5) and the cap (40000) would strengthen the claim that EAD is a robust, "plug-and-play" method.
>
> **A3**:  Thanks for bringing up this concern. In the revised manuscript, we’ve added additional ablation studies on the sensitivity of hyperparameters. Results are presented in Section 5.5 and Figure 11. Main conclusions are: (1) this global-step-aware decay rate is fairly robust once it’s not too small ($d_\max\ge200$), and (2) a small initial decay rate $d_0$ with a positive growth factor $\alpha$ is most beneficial to learning efficiency.
> To further examine the sensitivity of the global-step-aware decay rate, we conducted a stress test using a more challenging and complex setting: DAPO-recipe (trained on DAPO-17K dataset and validated on AIME-24 with Qwen-3-8B-Base as the base model) (See Section 5.6). In this case, the model needs to try really hard to solve the problem, and we observe that response length grows rapidly during training. To accommodate this fast growth, a larger growth factor proves to be beneficial for steady and stable training (See Figure 11).

---

### Official Review · Reviewer_aBdm · 2025-11-02

**Soundness:** 3
**Presentation:** 3
**Contribution:** 2
**Rating:** 6
**Confidence:** 2

**Summary:**

This paper introduces a new method of llm decoding that uses a high temperature for the token at the beginning of the sequence and gradually decay the temperature towards the end. The authors show that this method empirically improves inference time scaling, and can be incorporated into multiple RL algorithms to see further gains.

**Strengths:**

1. The method proposed is simple and can be easily integrated with current machine learning algorithms.

2. The empirical observation that exploring at the beginning phase is better than exploring at the later phase is insightful.

3. The paper provides comprehensive experiments to show the effectiveness of the method. They also provide empirical data related to the randomness to justify the statistical significance

4. The authors clearly noted that in some cases, their algorithms' advantage may diminish / become smaller. This significantly improve the clearity of the result.

**Weaknesses:**

1. While it is plausible that the inequality behind 172-174 approximately holds, it is not a consequence of data processing inequality unless there are strong assumptions.

2. [Minor] The arragement of Figure 9 and 10 brings some confusions as Figure 9 is positioned at the section discussing incorporation with RL algorithms.

**Questions:**

Please refer to the weaknesses. Some additional questions include

1. Why EAD seems to have a collapsing worst-of-all in Figure 4 for llama model and at the same time has the highest pass@16? What is the intuition behind this?

2. How is the global-step-aware decay rate decided and is it strictly required to sample the trained model from the RL process using the same EAD temperature schedule as it is trained?

---

> ### Author Response · Authors · 2025-11-21
>
> We thank Reviewer aBdm for their constructive feedback and for recognizing the simplicity, insightfulness, and effectiveness of our proposed EAD method.
>
> > **W1**: While it is plausible that the inequality behind 172-174 approximately holds, it is not a consequence of data processing inequality unless there are strong assumptions.
>
> **R1**: We acknowledge that the entropy reduction observed in Figure 1 is not a strict mathematical derivation from the Data Processing Inequality (DPI). Rather, we reference the DPI as a theoretical motivation to investigate entropy dynamics—specifically, to hypothesize where high- and low-entropy tokens are likely to reside. Our empirical verification confirms that high-entropy tokens generally appear at the beginning of generation, which aligns with the intuition provided by the DPI. We have revised the text to explicitly clarify that this connection serves as a motivational framework rather than a formal proof.
>
> > **W2**: [Minor] The arrangement of Figure 9 and 10 brings some confusions as Figure 9 is positioned at the section discussing incorporation with RL algorithms.
>
> **R2**: Thank you for pointing this out. We have adjusted the layout in the revision to ensure Figure 9 and 10 are positioned adjacent to their relevant textual discussions.
>
> > **Q1**: Why EAD seems to have a collapsing worst-of-all in Figure 4 for llama model and at the same time has the highest pass@16? What is the intuition behind this?
>
> **A1**: This phenomenon illustrates the trade-off between aggressive exploration and worst-case performance during training. The high Pass@16 indicates that EAD effectively encourages the model to explore a wider range of reasoning paths at the beginning of generation, which significantly increases the probability of discovering correct solutions.
> However, this broad exploration inevitably causes some trajectories to diverge far from the correct reasoning path. As the temperature anneals, these incorrect paths become "locked in," resulting in low-quality outliers (the collapsing worst-of-all). This is further validated by the TIS variant, where importance sampling balances high- and low-quality rollouts, mitigating this collapse. Crucially, we argue that lower worst-case performance during RL training is an acceptable cost for exploration; the model requires diverse signals—including failures—to improve its overall reasoning capabilities.
>
> > **Q2**: How is the global-step-aware decay rate decided and is it strictly required to sample the trained model from the RL process using the same EAD temperature schedule as it is trained?
>
> **A2**: Thank you for this excellent question. We address both parts below:
>
> 1. **Deciding the decay rate**: The decay rate is selected based on hyperparameter tuning on a held-out validation set. Our sensitivity analysis indicates that while strict tuning is not critical for the standard "Minimal-RL" recipe (please see results referenced in R2, Comment 3), the dynamic temperature schedule helps stabilize performance in more challenging training settings, such as DAPO-16k on AIME-24. In these harder tasks, the schedule aids the model in learning longer reasoning chains. We have added these ablation studies to the revision.
>
> 2. **Inference vs. Training**: The EAD temperature schedule is not required during inference. EAD is designed specifically to incentivize exploration during the RL training phase. In our experiments, we used the same fixed-temperature sampling for all methods during evaluation to ensure a fair comparison, and the EAD-trained model consistently outperformed baselines.
>
> 3. **Sensitivity Note**: In general, EAD is robust across different tasks and is not overly sensitive to this parameter. We find that decay rates between 25 and 2000 are effective, with larger rates (e.g., 500) benefiting harder tasks like AIME-24, while lower rates (e.g., 25) suffice for simpler tasks like MATH-500.

---

> > ### Comment · Reviewer_aBdm · 2025-11-26
> >
> > I would like to thank the authors for the detailed rebuttal and I will keep my positive rating.

---

> > > ### Author Response · Authors · 2025-11-26
> > > **Thanks for the comments!**
> > >
> > > We sincerely appreciate the thoughtful comments and constructive feedback from you. We hope our responses have adequately addressed your questions and concerns. As the discussion period deadline approaches, we look forward to your further feedback on our responses and would be happy to clarify any additional questions you may have.

---

### Author Response · Authors · 2025-11-21
**General Response**

We sincerely thank all reviewers for their constructive feedback. All reviewers agree that our paper is well-written with comprehensive empirical and ablation studies to show the effectiveness of the method. Also, both Reviewers aBdm and fG7W agree that our method is well-motivated and simple. For common concerns, we’ve conducted additional experiments and revised the paper to improve the clarity and completeness of our manuscript. We summarize our justification and revision as follows:

1. Reviewers fG7W and xBao mentioned related works adopting branching strategies or other training loss (designed for pass@k). We argue that our method is orthogonal to those methods: it is easy to use, and flexible to be integrated with any branching strategy or training loss. Our goal is to further encourage exploration for any baseline, and we encourage the use of our EAD together with any preferred algorithm to achieve better performance. We thank the reviewer for pointing out these related works, and we’ve included them in the latest version.

2. Reviewers aBdm and fG7W raised the concern about the sensitivity of the global-step-aware decay rate. We’ve conducted additional ablation studies and included the results in Figures 10 and 11 of the revised manuscript. The main conclusion is that **(1) this global-step-aware decay rate is fairly robust (does not require extensive tuning)**, and **(2) adopting it could be helpful for stabilizing training, especially when the response lengths grow rapidly during training (for hard tasks)**. More specifically,

    - For Math-500 experiments, we’ve added an ablation study using a fixed decay rate (independent of the global step). As shown in Figure 11(a), there is no significant difference in performance with or without decay rate adaptation during training. When adopting the global-step-aware decay rate, a slight improvement is observed.

    - We’ve further conducted a stress test under a more challenging setting: DAPO-recipe (trained on DAPO-17K dataset and validated on AIME-24 with Qwen-3-8B-Base as the base model). We see our proposed EAD significantly outperforms temperature sampling baselines. As the response length increases rapidly during training, a global-step-aware decay rate proves beneficial for steady and stable training. Results are shown in Figure 12 and discussed in Section 5.6.

In addition to these, we’ve performed an additional ablation study to illustrate the dynamic nature of EAD by comparing it against a “reverse annealing” decoding strategy. Figure 10 shows both the accuracy and the entropy domination of our EAD. We have also revised the manuscript to address potential misunderstandings noted by the reviewer. All revisions are highlighted in blue.

---

### Meta-Review · Area_Chair_5pCd · 2026-01-01

**Summary:**

Based on the insight that the early tokens in a sequence dominate the semantic direction and are most exploratory, this paper proposes exploratory annealed decoding for verifiable reinforcement learning, which anneals the sampling temperature from high to low during generation, to achieve a good balance between exploration diversity and sample quality. Experiments with multiple RLVR algorithms and models are conducted to show the improved sample efficiency and performance of the proposed method.

The major concern of this work lies in that the innovation is technically simple and the applicability may be limited due to the strong assumptions behind the proposed method.

**Reviewer Concerns:**

After reading the paper and all the discussion comments, the AC thinks the authors have addressed most of the reviewers' questions and concerns. However, the major concerns regarding the technical simplicity and the over-generalized assumption have not been well addressed. Although the authors reiterated the novelty and value of the proposed method in the rebuttal, the strong assumption – only early tokens are associated with exploration – still largely limits the applicability of the method because it does not hold universally, especially under complex reasoning tasks where discovery may happen late in a sequence. Therefore, the paper needs major improvements.

**Reviewer Scores:**

The final scores would be 2 (fG7W), 4 (xBao), 6 (aBdm).

---

### Decision · Program_Chairs · 2026-01-26

Reject